# Remote Sensing Scene Image Classification Based on mmsCNN–HMM with Stacking Ensemble Model

**Xiang Cheng** [1,2] and **Hong Lei** [1,*]

1 Department of Space Microwave Remote Sensing System, Aerospace Information Research Institute, Chinese Academy of Sciences, Beijing 100190, China

2 School of Electronic, Electrical and Communication Engineering, University of Chinese Academy of Sciences, Beijing 100039, China

* Correspondence: hlei@mail.ie.ac.cn

**Abstract:** The development of convolution neural networks (CNNs) has become a significant means to solve the problem of remote sensing scene image classification. However, well-performing CNNs generally have high complexity and are prone to overfitting. To handle the above problem, we present a new classification approach using an mmsCNN–HMM combined model with stacking ensemble mechanism in this paper. First of all, a modified multi-scale convolution neural network (mmsCNN) is proposed to extract multi-scale structural features, which has a lightweight structure and can avoid high computational complexity. Then, we utilize a hidden Markov model (HMM) to mine the context information of the extracted features of the whole sample image. For different categories of scene images, the corresponding HMM is trained and all the trained HMMs form an HMM group. In addition, our approach is based on a stacking ensemble learning scheme, in which the preliminary predicted values generated by the HMM group are used in an extreme gradient boosting (XGBoost) model to generate the final prediction. This stacking ensemble learning mechanism integrates multiple models to make decisions together, which can effectively prevent overfitting while ensuring accuracy. Finally, the trained XGBoost model conducts the scene category prediction. In this paper, the six most widely used remote sensing scene datasets, UCM, RSSCN, SIRI-WHU, WHU-RS, AID, and NWPU, are selected to carry out all kinds of experiments. The numerical experiments verify that the proposed approach shows more important advantages than the advanced approaches.

**Keywords:** remote sensing scene image classification; deep learning; CNN; hidden Markov model (HMM)

## 1. Introduction

Remote sensing image analysis is the understanding and research of surface semantic content. For the past few years, a great deal of remote sensing images with extremely high-quality clarity are easier to obtain, which promotes the development of many studies, such as remote sensing scene image classification [1], geographic image retrieval [2], and automatic target recognition [3]. As a significant topic, remote sensing image classification uses the computer to analyze various ground objects in the remote sensing image, select the features, and then label the category of the given image. Unlike ordinary images, remote sensing images are more difficult to process. For instance, remote sensing images include all kinds of classes of objects, which vary in scale, tint, and position. Apart from that, due to the interference of external factors during the collection process of remote sensing images, there are large intraclass differences and interclass similarities between them [4]. For example, as can be seen in Figure 1, a school area may be composed of various geographical structures, including playgrounds, baseball field, and squares, while they belong to disparate categories. All of these make it hard to accurately classify remote sensing scene images.

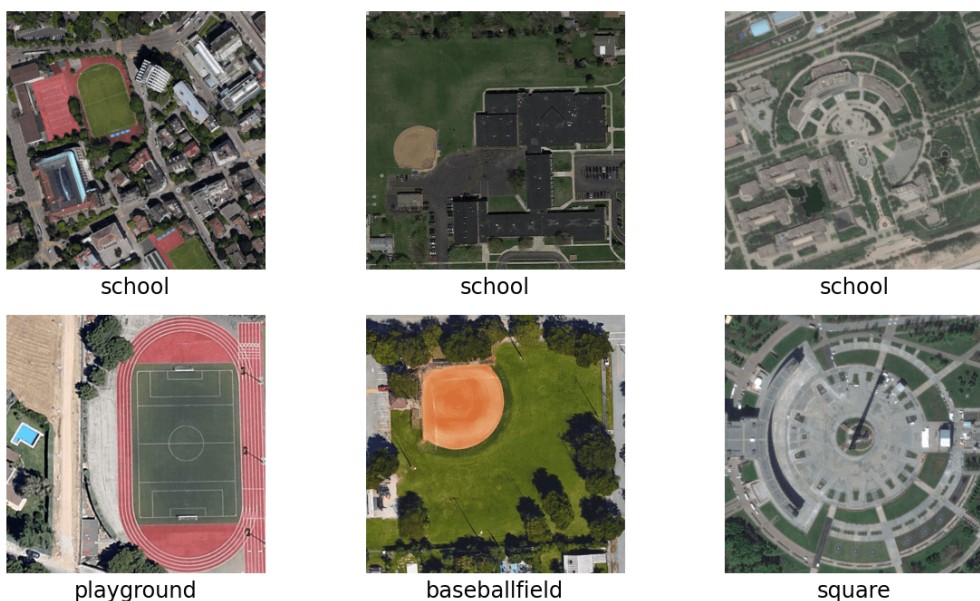

**Figure 1.** Several sample images from the AID datasets [5].

Presently, with the development of artificial intelligence, deep learning has become an absolutely important tool, and has been applied in many aspects, including image recognition [6,7] and speech enhancement [8]. In particular, CNN is one of the commonly used approaches for image classification on account of its vigorous feature extraction ability [9–11]. Many neural networks with different structures have been proposed for remote sensing scene image classification over the past few years [12–14]. For instance, ref. [15] proposed an efficient combined structure to take advantage of the strengths of CNN and CapsNet. Liang introduced a new two-stream architecture that combines CNN and graph convolutional network (GCN) [16]. In addition, ref. [4] proposed a self-compensated CNN to improve the speed of the model with low computational complexity. However, the performance of traditional CNNs in remote sensing scene image classification is not satisfactory.

To improve the performance of the CNN model in complex environments, it is of great significance to extract rich and representative features. Therefore, the multi-scale convolution neural network (msCNN) was proposed. At present, multi-scale convolutional neural networks can be divided into two types. The first is multi-scale convolutional neural network based on multi-scale images. This type of model usually inputs images of different scales into the same network model to obtain the features of images of different scales, and then fuses them to obtain multi-scale features. Researchers obtain multi-scale images in different ways. For example, the original image was transformed by Laplace pyramid in [17], and ref. [18] directly scaled the original image at different scales. Although msCNN based on multi-scale images has achieved good experimental results, its network structure consumes a lot of memory and cannot adapt to larger and deeper networks. The second is a convolutional neural network model based on multi-scale feature maps, such as the single-shot multibox detector (SSD) network in target detection tasks [19]. The SSD network uses multiple feature maps to perform location regression and classification, which can handle the problem of poor detection results caused by small objects on large-scale feature maps. Compared with the first type, the msCNN models based on multi-scale feature maps generally have lower memory requirements. However, due to the introduction of convolution kernels of various sizes, the number of parameters of the second msCNN explodes, which is more likely to lead to the gradient vanishing problem.

As is well known to us all, HMMs are usually used to model one-dimensional data. Recently, they have been used in computer vision, including texture segmentation [20], face finding [21], object recognition [22], and face recognition [23]. Specifically, ref. [24]

gave a detailed and in-depth introduction to HMMs. The ability to extract signal context information makes HMMs perform better than some traditional methods in the above applications. However, the effect of using the HMM alone is inferior to the current popular deep learning method. Therefore, we consider combining an HMM with a convolutional neural network to make full use of its feature extraction ability.

Ensemble learning is a technology that uses a variety of compatible learning algorithms to perform a single task in order to obtain better prediction performance [25]. To be specific, ensemble learning first uses the basic learners to extract a set of features and perform various transformations. Based on these learned features, a variety of basic learners produce preliminary prediction results. Finally, the top-level learner fuses the information from the above results and achieves better prediction performance through an adaptive voting scheme. Currently, common types of ensemble learning include bootstrap aggregating (bagging) [26], boosting [27], Bayesian model combination [28], and stacking [29]. As the most efficient ensemble learning method that can effectively resist overfitting, stacking is selected in this paper. In addition, HMMs in the HMM group are natural basic learners, which is consistent with the stacking framework in the ensemble.

In this paper, we design an mmsCNN–HMM combined model with stacking ensemble scheme, where the multi-scale features are extracted by mmsCNN, while the context information of the previously extracted features is mined by the hidden Markov model. The main contributions are as follows:

- A new framework of an mmsCNN–HMM combined model with stacking ensemble mechanism is presented.
- In view of the large computational complexity of the existing msCNN, we introduce the mechanism of shortcut connections, deriving from the residual network, into msCNN and propose a modified msCNN (mmsCNN). The mmsCNN model can effectively extract multi-scale structure features with less computational complexity, and avoid the gradient vanishing problem.
- An appropriate HMM is designed to mine the context information of the extracted features by mmsCNN, which can obtain abundant hidden structural feature information. For different categories of scene images, the corresponding HMM is trained and all the trained HMMs form an HMM group. Then, for each sample, the trained HMM group can give the preliminary prediction result.
- The proposed approach is based on a stacking ensemble learning scheme, in which the prediction results generated by the trained HMM group are used by the XGBoost model to generate the final prediction. This stacking ensemble learning scheme uses HMMs as natural basic learners, which could take advantage of the feature extraction ability of HMMs and prevent overfitting while ensuring accuracy.

The rest of this article is organized as follows. Section 2 gives a detailed introduction of the modified multi-scale convolution neural network, hidden Markov model, and stacking ensemble mechanism. In Section 3, we make several numerical experiments and comparisons on six remote sensing scene datasets, i.e., UCM [30], RSSCN [31], SIRI-WHU [32], WHU-RS [33], AID [5], and NWPU [34]. Finally, Section 4 summarizes our work and discusses our plans for the future.

Before the discussion, the mathematical symbols in this paper are explained. Boldface letters denote matrices or vectors. Lowercase letters denote scalars. In addition, the following mathematical symbols are used:

- $Conv[(m,n), i, j]$: convolution neural networks with convolution kernel size = $m \times n$, padding = $i$, and stride = $j$.
- $Pooling[(m,n), i, j]$: pooling layers with filer size = $m \times n$, padding = $i$, and stride = $j$.

## 2. Methodology

The processing flowchart of our model, i.e., an mmsCNN–HMM combined model with stacking ensemble mechanism, is shown in Figure 2. It consists of the following three

parts: the modified msCNN (mmsCNN), hidden Markov model (HMM), and the stacking ensemble mechanism.

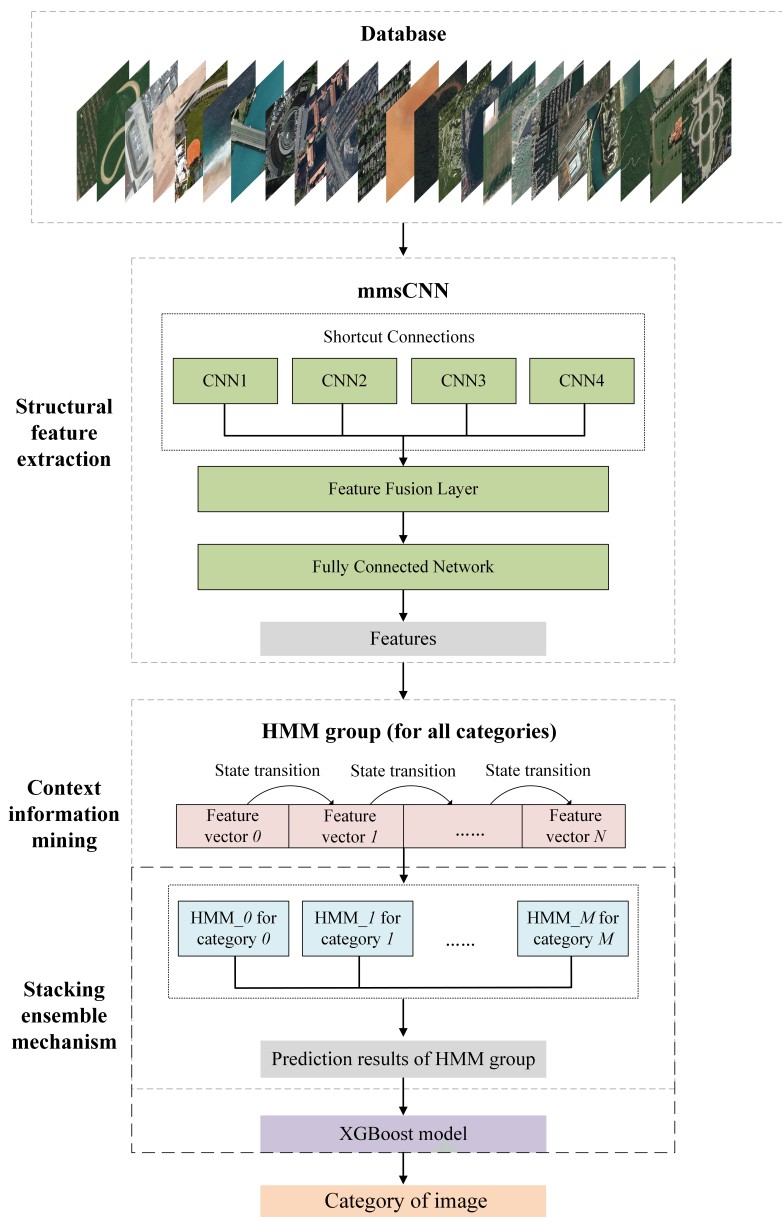

**Figure 2.** The basic framework of our model.

### 2.1. mmsCNN

Presently, multi-scale convolutional neural networks (msCNNs) mainly include two different types. The first is the msCNN model, on the basis of multi-scale images. This type of model usually inputs images of different scales into the same network model to obtain the features of images of different scales, and then fuses them to obtain multi-scale features. The second is the msCNN model based on multi-scale feature maps, which utilizes convolution kernels of different sizes to process images of the same size. Although msCNN based on multi-scale images has achieved good experimental results, its network structure consumes a lot of memory and cannot adapt to larger and deeper networks. Additionally, the existing msCNN models based on multi-scale feature maps are not perfect either, which introduces convolution kernels of various sizes, causing an explosion in the number of parameters. Therefore, we present a lightweight msCNN model based on multi-scale

feature maps to extract multi-scale features with fewer parameters and less calculation, as shown in Figure 3.

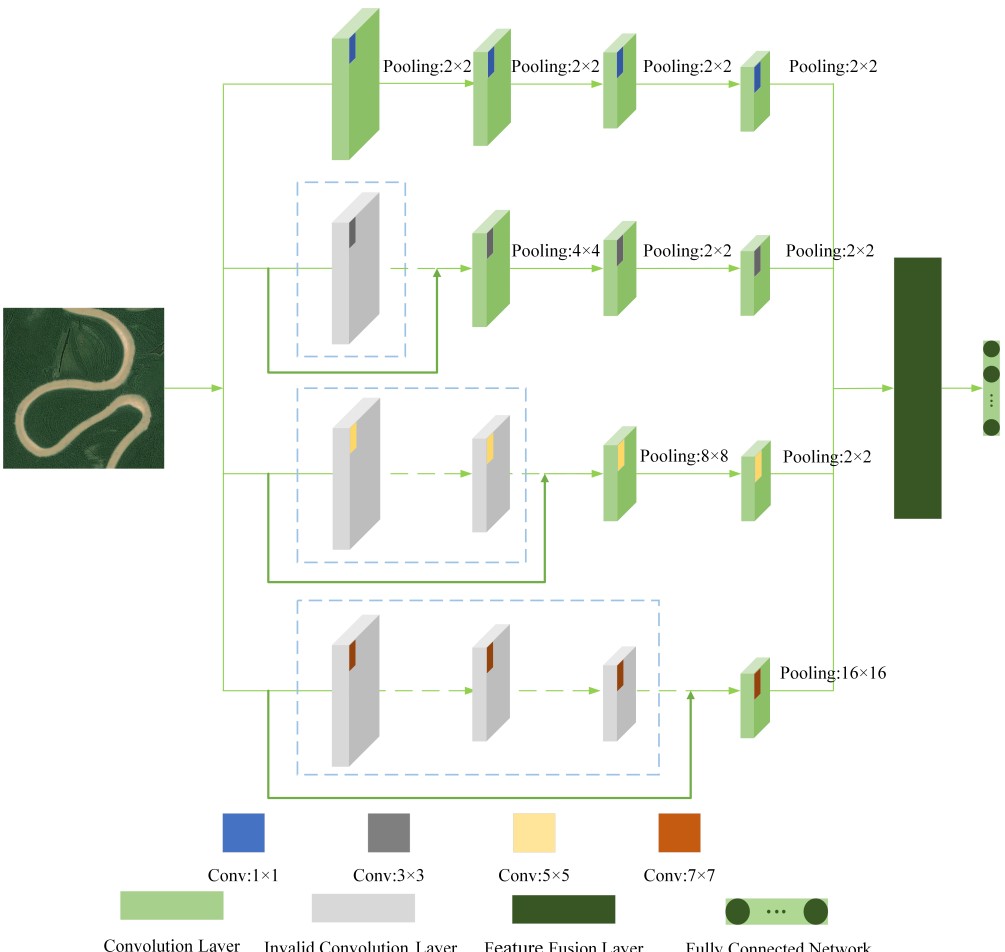

**Figure 3.** The specific structure of the modified msCNN (mmsCNN) model.

Recently, ref. [35] showed that the shallow large-kernel CNNs have a larger effective receptive field and are more consistent with the human perception mechanism. In addition, since the large-kernel CNNs need to update more parameters, the gradient vanishing is more likely to occur when the depth is as deep as the small-scale convolution kernel. The above two cases inspire us to introduce the mechanism of shortcut connections into large-kernel CNNs, which are derived from the residual network. First of all, we input the sample images into four different kernel CNNs, i.e., $Conv[(1, 1), 0, 1]$, $Conv[(3, 3), 1, 1]$, $Conv[(5, 5), 2, 1]$, and $Conv[(7, 7), 3, 1]$. Unlike the traditional msCNN based on multi-scale feature maps, we perform shortcut connections on the CNNs above. Specifically, we, respectively, invalidate the first 1, 2, and 3 convolution layers of $Conv[(3, 3), 1, 1]$, $Conv[(5, 5), 2, 1]$, and $Conv[(7, 7), 3, 1]$, as shown in the blue dotted box in Figure 3. In the following description, we refer to the network in the blue dashed box in Figure 3 as the invalid layer, which is exactly the network we skipped. To ensure the same size of different convolution layers during fusion, we adjust the pooling scale in the network of the next layer of the invalid convolution layer. Concretely speaking, the pooling layer after the invalid layer in $Conv[(3, 3), 1, 1]$ is set to $Pooling[(4, 4), 0, 4]$, that after the invalid layer in $Conv[(5, 5), 2, 1]$ is set to $Pooling[(8, 8), 0, 8]$, and that after the invalid layer in $Conv[(7, 7), 3, 1]$ is set to $Pooling[(16, 16), 0, 16]$. Obviously, the number of layers of large-kernel CNNs and parameters in the model are greatly reduced in the mmsCNN. Numerical experiments verify that the proposed mmsCNN can effectively extract multi-scale features with less computational complexity, and prevent the occurrence of gradient vanishing.

Next, multi-scale features are merged at the feature fusion layer. Since the size of features with different scales is consistent in the previous processing, these features can be added directly in the feature fusion layer. After feature merging, the output of the model contains both the signal details of the high-frequency features extracted from the small-scale feature maps and the low-frequency local feature information extracted from the large-scale feature maps. At the end of the mmsCNN, we utilize the fully connected network to synthesize the previously extracted features. Notably, the export of the mmsCNN is the multi-scale feature sequence and will be fed into the subsequent HMM as the observation sequence.

### 2.2. HMM

A hidden Markov model (HMM) is a machine learning model used to describe a Markov process [24]. In this section, we utilize the HMM to further mine the context information (or "hidden state evolution laws") between the features extracted from the mmsCNN. Specifically speaking, we use the HMM to improve the semantic scores of highly correlated related features and weaken the semantic scores of unrelated features. Then, the HMM gives the preliminary prediction result of the sample, which is fed into XGBoost.

Firstly, we give a description of what the "context information" of HMM mining is. Each remote sensing scene image is composed of several salient regions according to the object composition logic. The hidden states of these salient regions show certain order laws, which can reflect the characteristics of object composition. Different objects have various compositional logics, so the salient regions of different objects have state evolution laws (or "context information") in different orders. As shown in Figure 4, "Beach" scene images can be roughly divided into three regions: "sandbeach", "wave", and "seawater". The "Pond" scene images can be roughly divided into three regions: "land", "lake", and "land". The "Port" scene images can be roughly divided into three regions: "land", "vessels", and "sea". The "Playground" scene images can be roughly divided into three regions: "arc rubber runaway", "soccer field", and "arc rubber runaway". There is a certain order law between those regions. For example, "wave" must be in between "sandbeach" and "seawater". This is the universal logic of objects such as "Beach", and this logic is reflected in the change of hidden state, as shown in Figure 5.

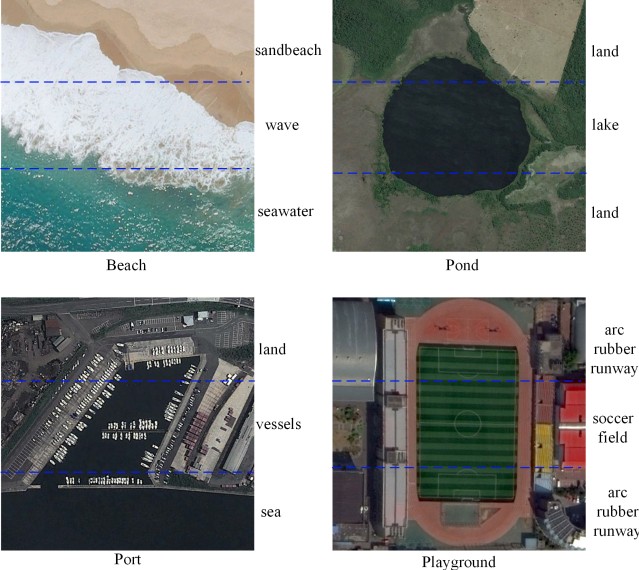

**Figure 4.** Three regions of different remote sensing images from the AID datasets [5].

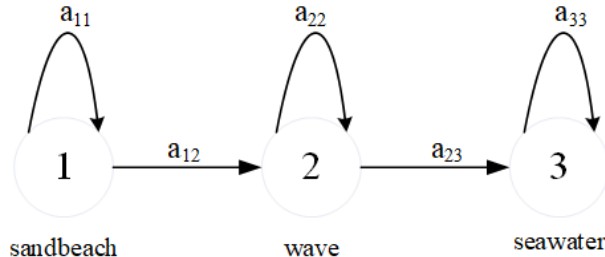

**Figure 5.** The HMM with three states for a "Beach" image.

What the HMM mines is the state evolution laws hidden behind features, and then it achieves the purpose of classification. Figure 6 shows the "hidden state evolution laws" or "context information" obtained by the HMM according to several different remote sensing images shown in Figure 4. Before designing the HMM, we use k-means to determine that there are eight hidden states for each event type signal. Figure 6a–d, respectively, show the hidden state evolution diagrams of three salient regions in the four scene images of "Beach", "Pond", "Port", and "Playground", which are obtained according to the Viterbi algorithm. As we can see, "Beach" and "Port" present three different hidden states, while the states of the first and third regions of "Pond" and "Playground" are the same, which is consistent with the image we observed.

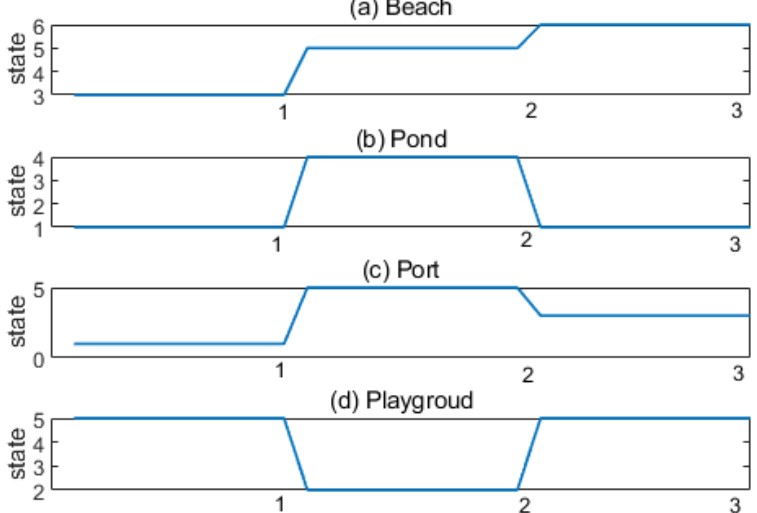

**Figure 6.** The "hidden state evolution laws" obtained by the HMM according to several different remote sensing images shown in Figure 4.

In this part, we discuss the importance of the features' priority exploited by the HMMs. In this paper, we first use the mmsCNN model to extract the features of each salient region in the image, and convert it into a feature sequence according to the order of the salient regions. Then, it is used as the observation sequence of the HMM to learn the internal state evolution law of the object, further excavate the composition logic of the object, and finally classify the image of the object. Therefore, the priority of features is important, and it needs to be arranged in the order of salient regions before it can be used to mine the changing laws of hidden states. In order to prove the importance of the priority of features, we use the "Beach" in Figure 4 as a sample to carry out comparative experiments. As shown in Figure 4, the sample contains three salient regions.

In this experiment, in the first step, the trained mmsCNN is used to extract a feature vector for the sample. In the second step, the mmsCNN feature vector, originally organized according to the channels, is rearranged as the feature sequences arranged in the order of the position of the salient regions of the sample. Specifically, this part of the work is mainly

to connect the features of all channels corresponding to the same position of the sample together as the features of the position. The mmsCNN feature sequence finally obtained in this step includes three subsequences, which correspond to the mmsCNN feature vectors of the three salient regions of the sample in turn. The third step is the most critical step, because the difference processing in this step constitutes the control of this experiment. The mmsCNN feature sequence output in the second step is directly used as the input of the control group, while the shuffled mmsCNN feature sequence is used as the input of the experimental group. To be specific, the processing of the experimental group in this step refers to exchanging the second subsequence and the third subsequence of the mmsCNN feature sequence. Although the internal order of each subsequence has not changed, the order of the exchanged mmsCNN feature sequence cannot be consistent with the position order of the salient region of the sample. In other words, after the exchange occurs, the features' priority of the feature sequence of the experimental group has been destroyed. In the fourth step, we input the feature sequence of the experimental group and the feature sequence of the control group into the HMM model corresponding to the trained "Beach" object. Then, we compare the prediction probabilities of the output of the HMM model in the experimental group and the control group, as shown in Table 1. In addition, the state evolution diagrams of the experimental group and the control group are shown in Figure 7.

**Table 1.** Comparison of two feature sequences in different order.

| Methods | The Probability of HMM (%) |
| --- | --- |
| The experimental group (the features' priority is destroyed) | 56.31 |
| The control group (the features' priority is retained) | **95.26** |

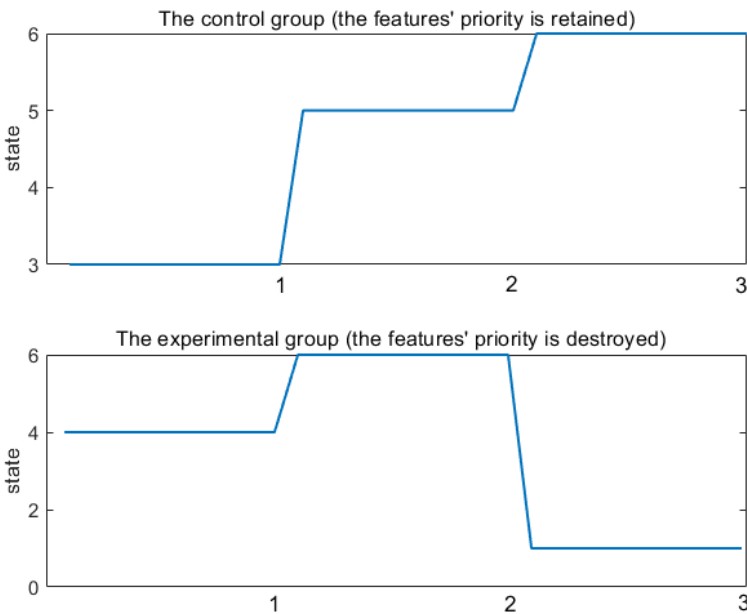

**Figure 7.** The "hidden state evolution laws" obtained by the HMM according to two different feature sequences.

As shown in Figure 7, compared with Figure 6a, there is a significant difference between the state evolution diagrams of the experimental group (the features' priority is destroyed) and the state evolution diagrams of the typical "Beach" sample, while there is no difference from the control group (the features' priority is retained). This indicates that when the features' priority in the observation sequence of HMM is destroyed, the object composition logic contained in the observation sequence changes markedly. As reported in Table 1, the probability of HMM output in the experimental group (features'

priority is destroyed) is significantly lower than that in the control group (features' priority is preserved). As we know, the higher the output probability, the more likelyit is that the HMM model thinks it belongs to this category. This proves that it is difficult for HMM to accurately classify according to the observation sequence when the features' priority in the observation sequence of HMM is destroyed. Based on the above two points, features' priority is extremely important for the HMM model to correctly match the internal structure logic of objects and thus correctly classify them.

Next, we explain how HMM works. The features output by the mmsCNN are fed into the HMM as the observation sequence, which are described as

$$\boldsymbol{o} = [o_1, o_2, \ldots, o_M], \tag{1}$$

assuming there are $M$ observation states in the Markov chain. In the training phase, the HMM parameters of each kind of image are initialized to

$$\boldsymbol{\lambda} = (\boldsymbol{\pi}, \boldsymbol{A}, \boldsymbol{B}). \tag{2}$$

Specifically, $\boldsymbol{\pi} = [\pi_n]$ is the original probability of each hidden state,

$$\pi_n = P(q_t = s_n), \quad 1 \le n \le N \tag{3}$$

where $N$ is the number of hidden states in the Markov chain and $q_t$ denotes the hidden state variable. In addition, $\boldsymbol{A} = [a_{nh}]_{N \times N}$ is denoted as the state transition probability matrix,

$$a_{nh} = P(q_{t+1} = s_h | q_t = s_n), \quad 1 \le n, h \le N \tag{4}$$

which describes the transition probability between states in HMM. $\boldsymbol{B} = (b_{nm})_{N \times M}$ is the probability matrix of the observation,

$$b_{nm} = P(p_t = o_m | q_t = s_n), \quad 1 \le n \le N, 1 \le m \le M \tag{5}$$

where $p_t$ denotes the observation variable. The matrix $\boldsymbol{B}$ is generally obtained by a Gaussian mixed model (GMM). Particularly, the more Gaussian models there are in a GMM, the better the effect of fitting the model, but the greater the computational complexity. The HMM structure is shown in Figure 6.

The main task in the training phase is to update parameter $\boldsymbol{\lambda}$ by using the feature vector output by mmsCNN. (We arrange the feature vectors extracted by mmsCNN into feature sequences in the order of salient regions. This step is mainly realized by flattening and rearranging the features of each channel). To be specific, $\boldsymbol{\lambda}$ is updated according to the Baum–Welch algorithm [36] to obtain the highest probability $P(\boldsymbol{o}|\boldsymbol{\lambda})$. We consider the probability $P(\boldsymbol{o}|\boldsymbol{\lambda})$ to reach its maximum value when the probability obtained in two consecutive iterations is almost equal. In the experiment, the convergence of training loss can be ensured after 10 iterations, which means that the difference of probability is close to $10^{-8}$. In summary, for a large number of observation sequences of each kind of image, we learn the context characteristics of this kind of image and obtain the parameters of the optimal HMM. Similarly, other HMMs for the scene images of other categories can be built in the same way. Ultimately, the HMMs of all kinds of images form an HMM group, which is used to discriminate all different scene images.

In the test phase, first of all, every image is transformed into a multi-scale feature sequence with the mmsCNN, which is then fed into the HMM group as a observation sequence. Subsequently, the HMMs in the HMM group score the matching degree according to the image features extracted by mmsCNN and the state evolution laws behind the features. In other words, we calculate the posteriori probability of observation sequence in each HMM on the basis of forward–backward algorithm [37], and then judge that the model with maximum probability is the image category. It is worth mentioning that if it is only feature matching or state evolution laws matching, the HMM will not obtain a high

probability value. Only when both the feature sequence and the state evolution law are matched with the parameters of the HMM can a high probability value be obtained and the scene category be determined. Figure 8 shows the recognition process of the HMM. Overall, for each sample image, the trained HMM group gives the preliminary prediction results and inputs them into XGBoost as training data.

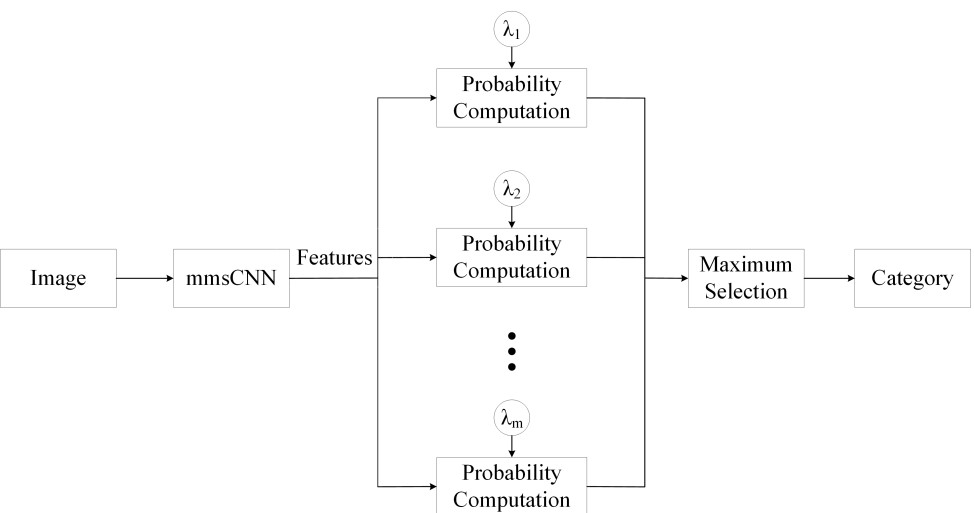

**Figure 8.** The recognition process of the HMM group.

At present, in addition to HMM, there are some other classification methods based on feature sequences, such as RNN. However, the RNN is black box since the hidden evolution routes of the signal are difficult to demonstrate, in which the improvement can only be seen from the final classification results, while in the HMM model, the sequential relationship can be also externally demonstrated, and it gives the reason why the model works. In addition, the use of the HMM for classification requires the construction of the HMM group, which naturally coincides with our stacking mechanism. The HMMs in the HMM group can be used as basic learners in the stacking mechanism to further improve the classification ability of the model; that is why we choose the HMM model at the beginning. Presently, we cannot prove that the performance of our model will be better or worse if the HMM is replaced with an RNN or LSTM model. The purpose of our work in this article is to use the HMM to find the state evolution laws of remote sensing image feature sequences and classify them. This is a start, and numerical experiments demonstrate that the effect of our model is considerable. In fact, one of our main tasks in the future is to focus on comparing the roles of HMM, RNN, and LSTM in our model, and find the best model.

### 2.3. Stacking Ensemble Mechanism

Ensemble learning is a technology to complete learning tasks by building and combining multiple learners [25]. At present, the three widely used ensemble learning methods are bagging, boosting and stacking. The core idea of bagging is to train a series of independent models in parallel, and then aggregate the output results of each model according to a certain strategy. The main idea of boosting is to train a series of dependent models serially. In other words, the latter model is used to correct the output results of the previous model. The main idea of stacking is to train a series of independent basic learners in parallel, and then combine the output results of each model by training a top-level learner.

In order to select the optimal ensemble learning scheme, we implement the above three ensemble learning methods on the UCM dataset. Presently, the bagging method and boosting method are widely accepted representative algorithms—random forest and XGBoost. Stacking is used as an algorithm framework, and researchers design the details of the top-level learner by themselves, with a higher degree of freedom. Therefore, we directly use mmsCNN–randomforest, mmsCNN–XGBoost and mmsCNN–stacking frameworks for

comparison. It is worth mentioning that the trained HMMs in the HMM group are used as the basic learners and XGBoost is used as the top-level learner to construct a single-layer stacking framework. The experimental results are shown in Table 2, which proves that higher classification accuracy can be obtained by using stacking, which is the reason why we choose stacking.

**Table 2.** Comparison of ensemble learning methods on the UCM dataset.

| Methods | OA (80/20) |
| --- | --- |
| mmsCNN–randomforest | $98.41 \pm 0.13$ |
| mmsCNN–XGBoost | $99.12 \pm 0.21$ |
| mmsCNN–stacking | $99.81 \pm 0.05$ |

In the stacking ensemble mechanism, the HMMs in the HMM group are natural basic learners and the top-level learner is the XGBoost model. If the HMM group consists of $m$ different HMMs in Section 2.2, the number of basic learners is $m$. Then, for each sample, the HMM group will obtain $m$ different probability values. Finally, these probability values and the label of the sample will be fed to the XGBoost model as training data, as shown in Figure 9. Different from traditional stacking ensemble mechanism, the proposed model naturally takes advantage of the differences of each HMM in the HMM group and does not rigidly link several models together, which can take advantage of the feature extraction characteristics of the HMMs in the HMM group. This stacking ensemble learning mechanism integrates multiple models to make decisions together, which can effectively avoid overfitting while ensuring accuracy. In addition, this stacking mechanism can handle several problems that basic learners cannot solve. For example, when two or more HMMs in the HMM group give the same highest score to the sample picture, the HMM group cannot judge the category of the sample picture. In this case, the trained XGBoost model in the stacking mechanism can still work.

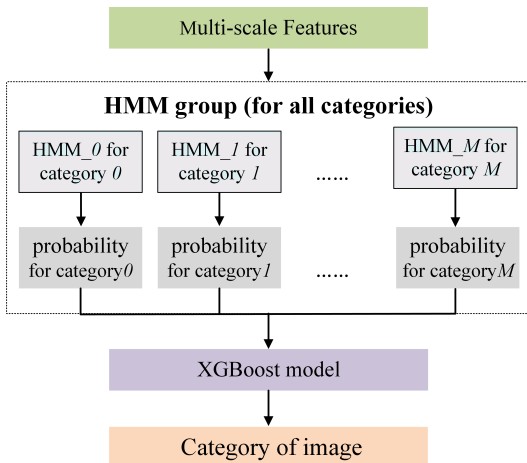

**Figure 9.** The process of the stacking ensemble mechanism.

In this paper, the XGBoost model is used as the top-level learner of the stacking framework, and the mmsCNN–HMM + XGBoost model is constructed. To demonstrate the validity of the XGBoost model, we performed ablation experiments on the UCM dataset. Specifically, we took the following two models as the control test of the mmsCNN–HMM + XGBoost model proposed in this paper: the first is mmsCNN–HMM, and the second is mmsCNN–HMM + SVM. As shown in Table 3, the experimental results prove that selecting the XGBoost model in stacking can result in higher classification accuracy.

**Table 3.** XGBoost ablation experiments on UCM dataset.

| Methods | OA (80/20) |
|---|---|
| mmsCNN–HMM | $97.21 \pm 0.17$ |
| mmsCNN–HMM + SVM | $98.92 \pm 0.21$ |
| mmsCNN–stacking | $99.81 \pm 0.05$ |

The basic idea of the extreme gradient boosting (XGBoost) model is to stack the outputs of diverse weak classification models to form a strong classification model. The stacking method adds the results of each weak classifier, i.e., classification and regression tree (CART), which is a typical binary decision tree. In the training phase, XGBoost trains the first tree with the training set to obtain the predicted value and the error with the sample truth value. Next, the second tree is trained, whose goal is to fit the residual of the first tree, and this procedure is the same as the first step except that the truth value is replaced by the residual of the first tree. After the second tree is trained, the residual of each sample can be obtained again, and then the third tree can be further trained, and so on. In the test phase, for each sample in the test set, each tree will have an output value, and the sum of these output values is the final predicted value of the sample. Then, the category of sample will be obtained on the basis of the final predicted value. Particularly, ref. [38] gave a detailed introduction to XGBoost.

## 3. Experiment

In this section, the six most widely used remote sensing scene datasets, UCM [30], RSSCN [31], SIRI-WHU [32], WHU-RS [33], AID [5], and NWPU [34], are selected to carry out all kinds of experiments. In addition, the proposed model is compared with several advanced approaches. To guarantee the validity of experiments, every model is performed using the same hyperparameters and equipment.

### 3.1. Datasets

In order to show the effect of our model, several experiments are carried out on the above six datasets. Table 4 reports the details of the six datasets. On account of the various image sizes in the datasets, in order to prevent memory overflow, we utilize the nearest neighbor interpolation approach to resize training images to $256 \times 256$. Figures 10 and 11 show several images in the UCM and AID datasets.

**Table 4.** Detailed introduction of six different datasets.

| Datasets | Number of Images per Class | Number of Classes | Number of Images in the Dataset | Spatial Resolution (m) | Images Size |
|---|---|---|---|---|---|
| UCM | 100 | 21 | 2100 | 0.3 | $256 \times 256$ |
| RSSCN | 400 | 7 | 2800 | - | $400 \times 400$ |
| AID | 200~400 | 30 | 10,000 | 0.5~0.8 | $600 \times 600$ |
| NWPU | 700 | 45 | 31,500 | 0.2~30 | $256 \times 256$ |
| WHU-RS | 50 | 19 | 1005 | 0.5 | $600 \times 600$ |
| SIRI-WHU | 200 | 12 | 2400 | 2 | $200 \times 200$ |

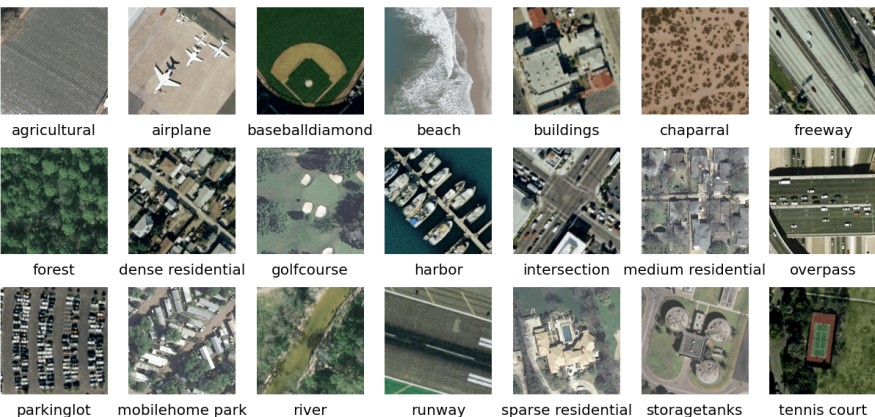

**Figure 10.** Several images in UCM dataset.

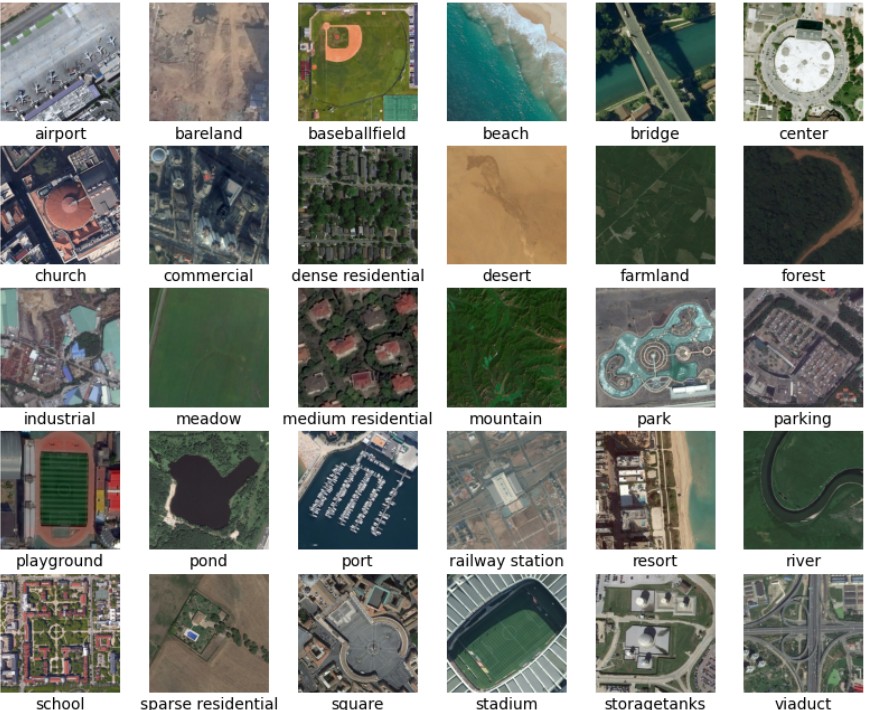

**Figure 11.** Several images in AID dataset.

### 3.2. Setting of the Experiments

In dataset partitioning, the stratified sampling method is used to avoid sampling bias. Specifically, we set a random seed such that the images are the same in all experiments. Apart from that, 20 independent repeated tests are conducted for all experiments, and the average results of 20 tests are taken as the final results. For the division of datasets, we follow the default guidelines as in other previous papers. Concretely, the UCM dataset is set to train:test = 8:2; the RSSCN dataset is set to train:test = 5:5; the SIRI-WHU dataset is set to train:test = 5:5 and train:test = 8:2; the WHU-RS dataset is set to train:test = 4:6 and train:test = 6:4; the AID dataset is set to train:test = 2:8 and train:test = 5:5; and the NWPU dataset is set to train:test = 1:9 and train:test = 2:8.

For the parameter setting of the mmsCNN, the learning rate is set to 0.01. Table 5 reports the structure parameters of the mmsCNN model. The batch size is 10 and the momentum during training is set to 0.9. For the parameter setting of HMM, $\lambda = (\pi, A, B)$ needs to be initialized. Firstly, we randomly initialize $\pi$ and $A$ under the condition that the sum of probability is equal to 1. As for $B$, Gaussian HMM is adopted and the number of Gaussian models in GMM is set to 8. In addition, the threshold value of the HMM is set to

$10^{-8}$, which means that when the difference of probability $P(o|\lambda)$ is equal to $10^{-8}$ in two adjacent iterations, the Baum–Welch algorithm stops iteration. For the parameter setting of XGBoost, the learning rate is initialized to 0.01. The maximum depth of the tree is 6 and the maximum number of iterations is initialized to 100. Additionally, the sampling rate of training samples when generating the next tree is set to 1.

**Table 5.** Structure parameters of the mmsCNN.

| Type | Number | Filter Size | Pad | Stride |
|---|---|---|---|---|
| Conv1 + ReLU | 64 | $1 \times 1$ | 0 | 1 |
| Max Pooling | - | $2 \times 2$ | 0 | 2 |
| Conv2 + ReLU | 128 | $(1/3) \times (1/3)$ | 0/1 | 1 |
| Max Pooling | - | $(2/4) \times (2/4)$ | 0 | 2/4 |
| Conv3 + ReLU | 256 | $(1/3/5) \times (1/3/5)$ | 0/1/2 | 1 |
| Max Pooling | - | $(2/4/8) \times (2/4/8)$ | 0 | 2/4/8 |
| Conv4 + ReLU | 512 | $(1/3/5/7) \times (1/3/5/7)$ | 0/1/2/3 | 1 |
| Max Pooling | - | $(2/4/8/16) \times (2/4/8/16)$ | 0 | 2/4/8/16 |
| Fully Connected Network + ReLU | 100 | $1 \times 1$ | - | - |

*3.3. Experimental Results*

In order to show the effect of our method, the overall accuracy (OA) and confusion matrix are selected as the evaluation indicators for comparison.

3.3.1. Experimental Results on the UCM Dataset

The first experiment is conducted on the UCM dataset. The methods from 2019 to 2021 are selected for comparison, and Table 6 reports the overall accuracies of them. Compared with the advanced classification methods, the proposed model achieved the highest accuracy of 99.81%, which is superior to all methods. Specifically, the accuracy of our method is 1% higher than the aggregated deep Fisher feature [39], 0.52% higher than the LCNN–BFF method [40], and 0.29% higher than LCNN–CMGF [41].

**Table 6.** OA comparison on UCM dataset.

| Methods | OA (80/20) | Year |
|---|---|---|
| Fine-Tune MobileNetV2 [42] | $98.13 \pm 0.33$ | 2019 |
| VGG-16-CapsNet [15] | $98.81 \pm 0.12$ | 2019 |
| AResNet + WSPM-CRC [43] | 97.95 | 2019 |
| Feature Aggregation CNN [44] | $98.81 \pm 0.24$ | 2019 |
| Aggregated Deep Fisher Feature [39] | $98.81 \pm 0.51$ | 2019 |
| LiG with Sigmoid Kernel [45] | $98.92 \pm 0.35$ | 2020 |
| Contourlet CNN [46] | $98.97 \pm 0.21$ | 2020 |
| MobileNet [47] | $96.33 \pm 0.15$ | 2020 |
| EfficientNet [48] | $94.37 \pm 0.14$ | 2020 |
| Positional Context Aggregation [49] | $99.21 \pm 0.18$ | 2020 |
| LCNN–BFF Method [40] | $99.29 \pm 0.24$ | 2020 |
| DDRL-AM Method [50] | $99.05 \pm 0.08$ | 2020 |
| Skip-Connected CNN [51] | $98.04 \pm 0.23$ | 2020 |
| Gated Bidirectiona + Global Feature [52] | $98.57 \pm 0.48$ | 2020 |
| HABFNet [53] | $99.29 \pm 0.35$ | 2020 |
| VGG-VD16 with SAFF [54] | $97.02 \pm 0.78$ | 2020 |
| EfficientNetB3-Attn-2 [55] | $99.21 \pm 0.22$ | 2021 |
| LCNN–CMGF [41] | $99.52 \pm 0.34$ | 2021 |
| Proposed | $\mathbf{99.81 \pm 0.05}$ | 2022 |

The confusion matrix of our model on the UCM dataset is shown in Figure 12. The accuracies of almost all categories reached 100%, except the classification accuracies of two

scenes of "dense residential" and "medium residential" which were 98%. It is likely that similar buildings in "dense residential" and "medium residential" scenes led to misclassification. The high classification accuracy of all scenarios verifies that our method has excellent performance on the UCM dataset.

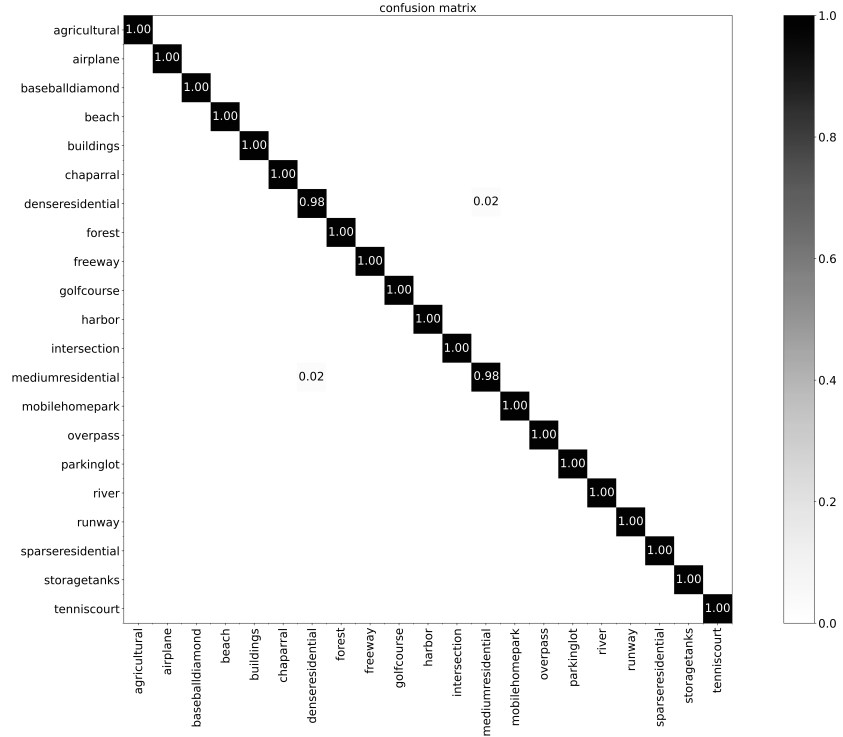

**Figure 12.** Confusion matrix of our model on UCM dataset.

### 3.3.2. Experimental Results on SIRI-WHU Dataset

The second experiment is conducted on the SIRI-WHU dataset with train:test = 5:5 and train:test = 8:2, and the overall accuracies of all kinds of methods are reported in Table 7. It is obvious that our method shows the best classification accuracy among all comparison methods. To be specific, when train:test = 5:5, the accuracy of the proposed method is 7.11% higher than DMTM [32], 5.19% higher than SRSCNN [56], 1.67% higher than SE-MDPMNet [42], and 0.55% higher than the SCCNN method [4]. When train:test = 8:2, the accuracy of the proposed method is 9.83% higher than LPCNN [57], 4.64% higher than pretrained-AlexNet-SPP-SS [58], 4.95% higher than SRSCNN [56], 0.94% higher than SE-MDPMNet [42], and 0.34% higher than the SCCNN method [4].

**Table 7.** OA comparison on SIRI-WHU dataset.

| Methods | OA (50/50) | OA (80/20) | Year |
|---|---|---|---|
| DMTM [32] | 91.52 | - | 2016 |
| LPCNN [57] | - | 89.88 | 2016 |
| SICNN [59] | - | 93.00 | 2016 |
| Pretrained-AlexNet-SPP-SS [58] | - | 95.07 ± 1.09 | 2017 |
| SRSCNN [56] | 93.44 | 94.76 | 2018 |
| Siamese ResNet-50 [60] | 95.75 | 97.50 | 2019 |
| Siamese AlexNet [60] | 83.25 | 88.96 | 2019 |
| Siamese VGG-16 [60] | 94.50 | 97.30 | 2019 |
| Fine-tune MobileNetV2 [42] | 95.77 ± 0.16 | 96.21 ± 0.31 | 2019 |
| SE-MDPMNet [42] | 96.96 ± 0.19 | 98.77 ± 0.19 | 2019 |
| The SCCNN Method [4] | 98.08 ± 0.45 | 99.37 ± 0.26 | 2021 |
| Proposed | **98.63 ± 0.15** | **99.71 ± 0.21** | 2022 |

The confusion matrix of our model on the SIRI-WHU dataset is shown in Figure 13. The accuracies of almost all categories reached 100%, except the classification accuracies of three scenes of "meadow", "residential", and "river". Several "residential" and "industrial" scenes are confused as they both include some buildings and grass, which leads to high class similarity. In addition, some "river" images are incorrectly classified as "water", which may be due to the fact that the "river" image contains water. However, our model still achieves high accuracy on SIRI-WHU dataset.

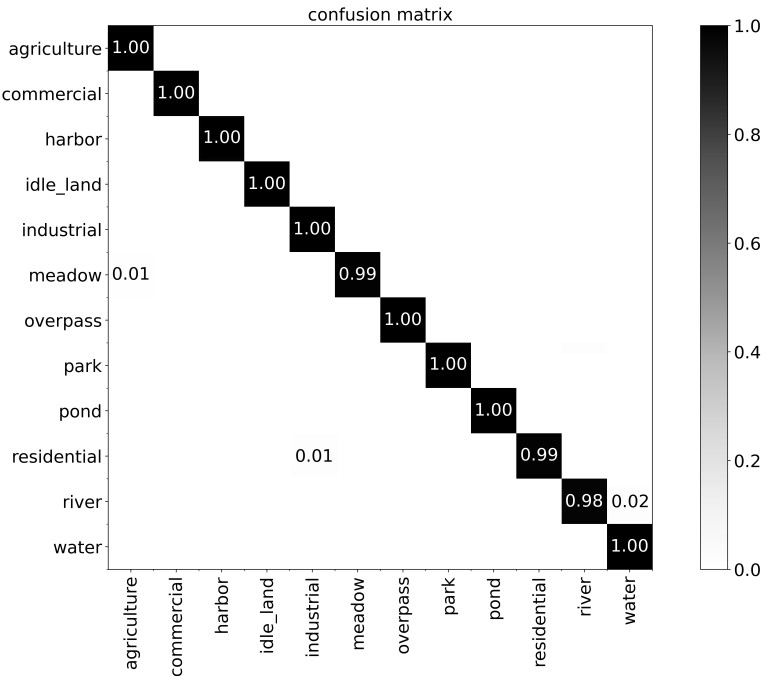

**Figure 13.** Confusion matrix of our model on SIRI-WHU dataset (train:test = 8:2).

### 3.3.3. Experimental Results on RSSCN Dataset

The third experiment is conducted on the RSSCN dataset with train:test = 5:5, and Table 8 reports the final results. Compared with the advanced models proposed recently, the proposed model in this paper shows the highest accuracy. Specifically, the accuracy of our model is 1.37% higher than VGG-16-CapsNet [15], 2.04% higher than positional context aggregation [49], and 0.52% higher than LCNN–CMGF [41].

**Table 8.** OA comparison on RSSCN dataset.

| Methods | OA (50/50) | Year |
| --- | --- | --- |
| VGG-16-CapsNet [15] | $96.65 \pm 0.23$ | 2019 |
| SPM-CRC [43] | 93.86 | 2019 |
| AResNet + WSPM-CRC [43] | 93.60 | 2019 |
| Aggregated Deep Fisher Feature [39] | $95.21 \pm 0.50$ | 2019 |
| SE-MDPMNet [42] | $92.46 \pm 0.66$ | 2019 |
| Variable-Weighted Multi-Fusion [61] | 89.1 | 2019 |
| Contourlet CNN [46] | $95.54 \pm 0.17$ | 2020 |
| Positional Context Aggregation [49] | $95.98 \pm 0.56$ | 2020 |
| LCNN–BFF Method [40] | $94.64 \pm 0.12$ | 2020 |
| LCNN–CMGF [41] | $97.50 \pm 0.21$ | 2021 |
| Proposed | $\mathbf{98.02 \pm 0.10}$ | 2022 |

The confusion matrix of our model on the RSSCN dataset is shown in Figure 14. As can be seen, the classification accuracies of all categories reach 97% or higher, and the accuracy of "Resident" can reach 100%. On account of the strong class similarity among

"Grass", "Forest", and "Field", the accuracy of "Field" is the lowest. Overall, Figure 14 demonstrates the effectiveness of our model on the RSSCN dataset.

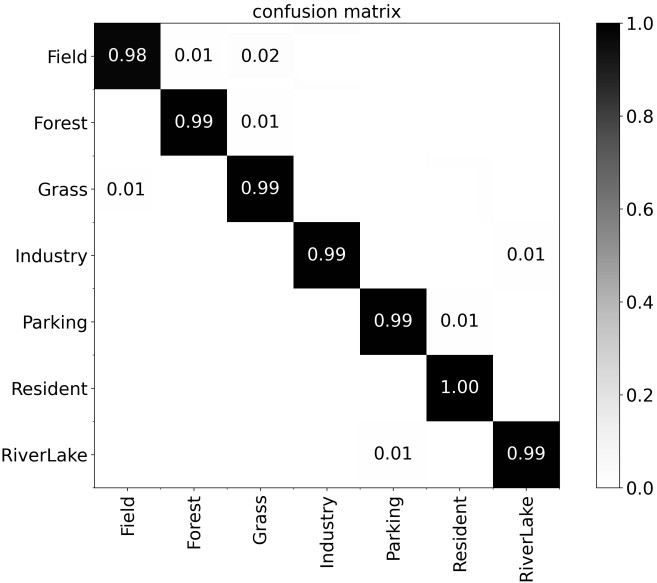

**Figure 14.** Confusion matrix of our model on RSSCN dataset.

3.3.4. Experimental Results on AID Dataset

The fourth experiment is performed on the AID dataset with train:test = 2:8 and train:test = 5:5, and Table 9 reports the experiment results. Compared with the advanced classification approaches, the proposed method shows more important advantages.

Specifically, when train:test = 2:8, the accuracy of our model is 1.15% higher than aggregated deep Fisher feature [39], 0.66% higher than InceptionV3 [62], and 0.30% higher than LCNN–CMGF [41]. When train:test = 5:5, the accuracy of the proposed method is 2.36% higher than feature aggregation CNN [44], 1.06% higher than SRSCNN [56], and 0.27% higher than LCNN–CMGF [41].

**Table 9.** OA comparison on AID dataset.

| Methods | OA (20/80) | OA (50/50) | Year |
|---|---|---|---|
| Feature Aggregation CNN [44] | - | $95.45 \pm 0.11$ | 2019 |
| Bidirectional Adaptive Feature Fusion [63] | - | 93.56 | 2019 |
| Aggregated Deep Fisher Feature [39] | $92.78 \pm 0.57$ | $95.26 \pm 0.84$ | 2019 |
| MobileNet [47] | $88.53 \pm 0.17$ | $90.91 \pm 0.18$ | 2020 |
| EfficientNet [48] | $86.56 \pm 0.17$ | $88.35 \pm 0.16$ | 2020 |
| InceptionV3 [62] | $93.27 \pm 0.17$ | $95.07 \pm 0.22$ | 2020 |
| MG-CAP with Bilinear [64] | $92.11 \pm 0.15$ | $95.14 \pm 0.12$ | 2020 |
| LCNN–BFF [40] | $92.06 \pm 0.36$ | $94.53 \pm 0.24$ | 2020 |
| DDRL-AM Method [50] | $91.56 \pm 0.49$ | $94.08 \pm 0.35$ | 2020 |
| GBNet [52] | $90.16 \pm 0.24$ | $93.72 \pm 0.34$ | 2020 |
| GBNet + Global Feature [52] | $92.20 \pm 0.23$ | $95.48 \pm 0.12$ | 2020 |
| HABFNet [53] | $93.01 \pm 0.43$ | $96.75 \pm 0.52$ | 2020 |
| VGG-VD16 with SAFF Method [54] | $92.05 \pm 0.34$ | $95.98 \pm 0.70$ | 2020 |
| ResNet50 [62] | $92.39 \pm 0.15$ | $94.69 \pm 0.19$ | 2020 |
| VGG19 [62] | $87.73 \pm 0.25$ | $91.71 \pm 0.24$ | 2020 |
| Skip-Connected CNN [51] | $91.10 \pm 0.15$ | $93.30 \pm 0.13$ | 2020 |
| EfficientNetB3-Attn-2 [55] | $92.48 \pm 0.76$ | $95.39 \pm 0.43$ | 2021 |
| LCNN–CMGF [41] | $93.63 \pm 0.10$ | $97.54 \pm 0.25$ | 2021 |
| Proposed | $\mathbf{93.93 \pm 0.15}$ | $\mathbf{97.81 \pm 0.04}$ | 2022 |

The confusion matrix of our method on the AID dataset is shown in Figure 15. The accuracies of all classes reached 90% or higher. Particularly, the results of "Desert", "Pond", "River", "Square", and "StorageTanks" reach 100%. The "School" scene shows the low classification accuracy of 94%. The reason is that "BaseballField", "Playground", and "Square" often appear in "School" scenes. Overall, Figure 15 shows the effectiveness of our model on this dataset.

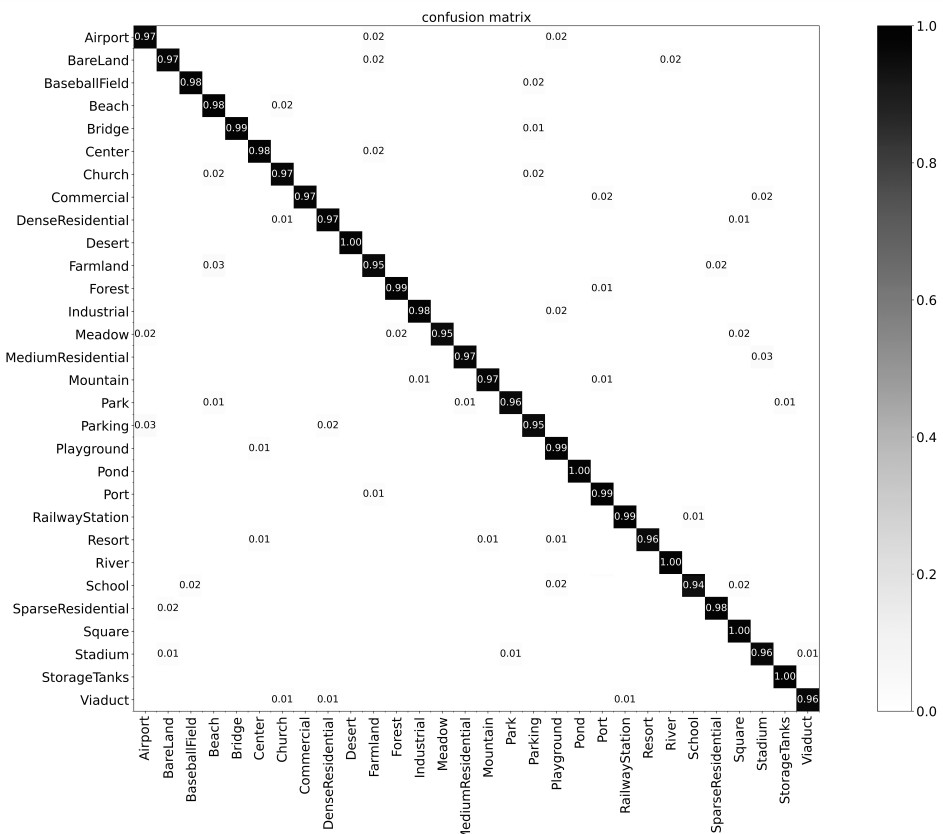

**Figure 15.** Confusion matrix of our method on AID dataset (train:test = 5:5).

### 3.3.5. Experimental Results on WHU-RS Dataset

The fifth experiment is performed on the WHU-RS dataset with train:test = 4:6 and train:test = 6:4, and Table 10 reports the experiment results. Compared with the other models, our model shows more important advantages. Concretely speaking, when train:test = 4:6, the accuracy of our model is 3.52% higher than VGG-VD-16 [5], 0.73% higher than two-stream deep fusion framework [65], 0.5% higher than SE-MDPMNet [42], 0.48% higher than TEX-Net-LF [52], and 0.31% higher than the SCCNN method [4]. When train:test = 6:4, the accuracy of our model is 1.03% higher than DCA by addition [66], 0.81% higher than two-stream deep fusion framework [65], 0.76% higher than SE-MDPMNet [42], 0.85% higher than TEX-Net-LF [52], and 0.22% higher than the SCCNN method [4].

The confusion matrix of our method on the WHU-RS dataset is shown in Figure 16. Obviously, most of the scene categories reach 100% except "forest" and "river". It is the large class similarity between "forest" and "river" that results in their low classification accuracy. For instance, a large number of trees may appear in "forest" and "river" scenes, which leads to the wrong classification of "river" as "forest". However, our method still achieves good classification results on the WHU-RS dataset.

**Table 10.** OA comparison on WHU-RS dataset.

| Models | OA (40/60) | OA (60/40) | Year |
|---|---|---|---|
| CaffeNet [5] | 95.11 ± 1.20 | 96.24 ± 0.56 | 2017 |
| VGG-VD-16 [5] | 95.44 ± 0.60 | 96.05 ± 0.91 | 2017 |
| GoogLeNet [5] | 93.12 ± 0.82 | 94.71 ± 1.33 | 2017 |
| DCA by addition [66] | - | 98.70 ± 0.22 | 2017 |
| Two-stream Deep Fusion Framework [65] | 98.23 ± 0.56 | 98.92 ± 0.52 | 2018 |
| Fine-tune MobileNetV2 [42] | 96.82 ± 0.35 | 98.14 ± 0.33 | 2019 |
| SE-MDPMNet [42] | 98.46 ± 0.21 | 98.97 ± 0.24 | 2019 |
| TEX-Net-LF [52] | 98.48 ± 0.37 | 98.88 ± 0.49 | 2020 |
| The SCCNN Method [4] | 98.65 ± 0.45 | 99.51 ± 0.15 | 2021 |
| Proposed | **98.96 ± 0.25** | **99.73 ± 0.27** | 2022 |

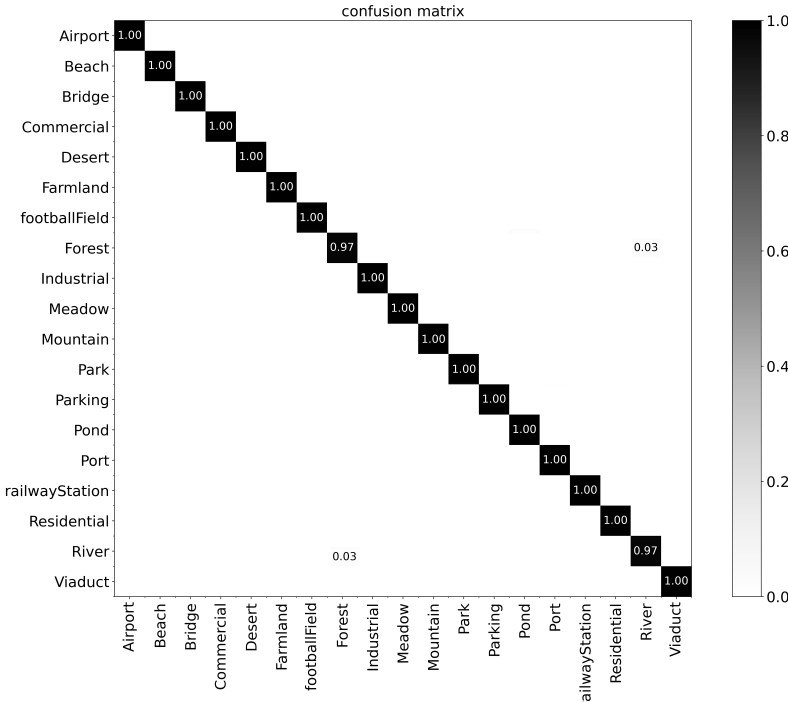

**Figure 16.** Confusion matrix of our model on WHU-RS dataset (train:test = 6:4).

### 3.3.6. Experimental Results on NWPU Dataset

The sixth experiment is performed on the NWPU dataset with train:test = 1:9 and train:test = 2:8, and the experiment results are reported in Table 11. It is obvious that our method shows the highest classification accuracy among all comparison methods. To be specific, when train:test = 1:9, the accuracy of our model is 4.21% higher than discriminative + VGG16 [67], 8.38% higher than VGG-16-CapsNet [15], 1.87% higher than MSDFF [68], and 0.90% higher than the SCCNN method andLCNN–CMGF [41]. When train:test = 2:8, the accuracy of our model is 3.62% higher than discriminative + VGG16 [67], 4.48% higher than R.D [69], 1.96% higher than MSDFF [68], and 1.12% higher than the SCCNN method [4].

The confusion matrix of the proposed model with the NWPU dataset is shown in Figure 17. As can be seen, the classification accuracies of most classes reach more than 90%. The lowest classification accuracies are "dense residential" and "medium residential", which reach 89% and 88%, respectively. The reason is that similar buildings exist in "dense residential", "medium residential", "tennis court", and "church" scenes, which results in large class similarity among them. Nevertheless, the proposed model still achieves excellent classification results on the NWPU dataset.

**Table 11.** OA comparison on NWPU dataset.

| Models | OA (10/90) | OA (20/80) | Year |
|---|---|---|---|
| Discriminative + VGG16 [67] | 89.22 ± 0.50 | 91.89 ± 0.22 | 2018 |
| Discriminative + AlexNet [67] | 85.56 ± 0.20 | 87.24 ± 0.12 | 2018 |
| VGG-16-CapsNet [15] | 85.05 ± 0.13 | 89.18 ± 0.14 | 2019 |
| R.D [69] | - | 91.03 | 2019 |
| Contourlet CNN [46] | 85.93 ± 0.51 | 89.57 ± 0.45 | 2020 |
| MG-CAP with Biliner [64] | 89.42 ± 0.19 | 91.72 ± 0.16 | 2020 |
| LiG with RBF Kernel [70] | 90.23 ± 0.13 | 93.25 ± 0.12 | 2020 |
| EfficientNet [48] | 78.57 ± 0.15 | 81.83 ± 0.15 | 2020 |
| LiG with Sigmoid Kernel [45] | 90.19 ± 0.11 | 93.21 ± 0.12 | 2020 |
| ResNet50 [62] | 81.34 ± 0.32 | 83.57 ± 0.37 | 2020 |
| VGG19 [62] | 86.23 ± 0.41 | 88.93 ± 0.12 | 2020 |
| InceptionV3 [62] | 85.46 ± 0.33 | 87.75 ± 0.43 | 2020 |
| MobileNet [47] | 80.32 ± 0.16 | 83.26 ± 0.17 | 2020 |
| Skip-Connected CNN [51] | 84.33 ± 0.19 | 87.30 ± 0.23 | 2020 |
| LCNN–BFF Method [40] | 86.53 ± 0.15 | 91.73 ± 0.17 | 2020 |
| VGG-VD16 with SAFF Method [54] | 84.38 ± 0.19 | 87.86 ± 0.14 | 2020 |
| MSDFF [68] | 91.56 | 93.55 | 2020 |
| LCNN–CMGF [41] | 92.53 ± 0.56 | 94.18 ± 0.35 | 2021 |
| The SCCNN Method [4] | 92.02 ± 0.50 | 94.39 ± 0.16 | 2021 |
| Proposed | **93.43 ± 0.25** | **95.51 ± 0.21** | 2022 |

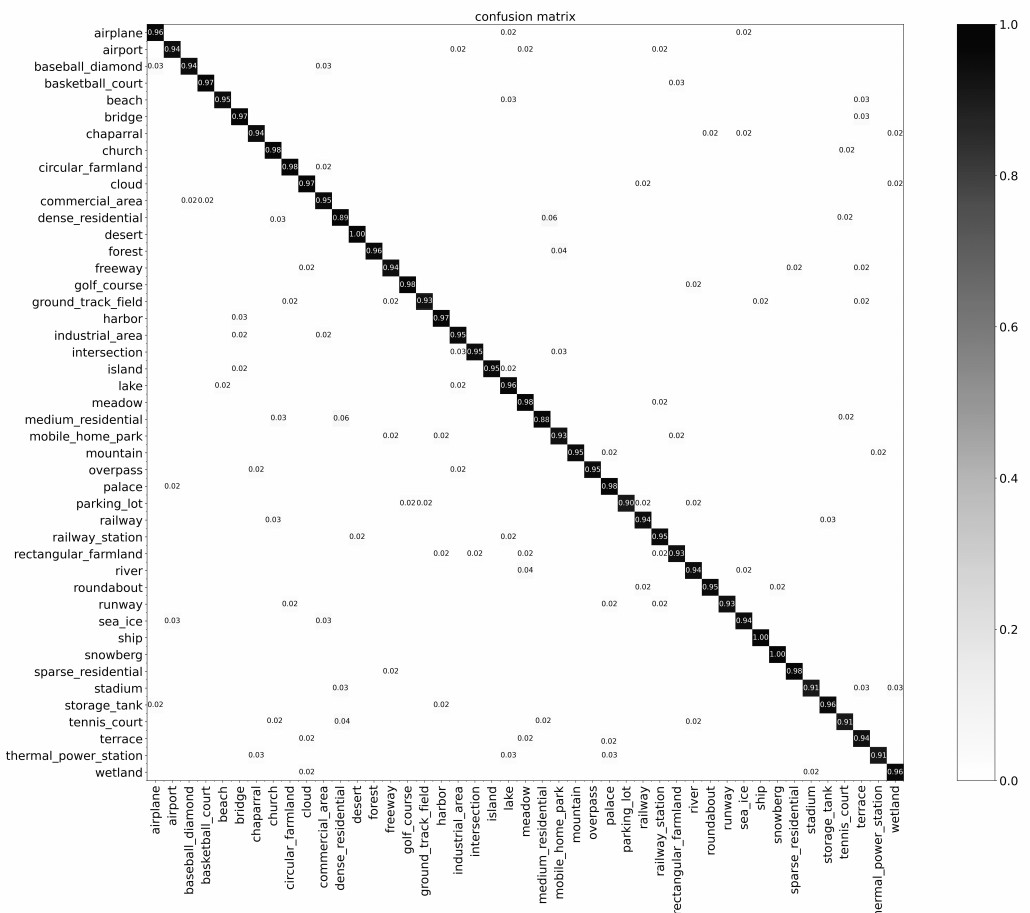

**Figure 17.** Confusion matrix of our model on NWPU dataset (train:test = 2:8).

### 3.3.7. Complexity Analysis

In this section, we compare the parameter quantity of our model with other models. Apart from that, floating point operations per second (FLOPs) are used to measure the time

complexity of the model. We give the parameter quantities and FLOPs of each model on the RSSCN dataset, as shown in Table 12.

**Table 12.** Complexity comparison on RSSCN dataset.

| Methods | OA (50/50) | Parameter Quantities | FLOPs |
|---|---|---|---|
| VGG-16-CapsNet [15] | $96.65 \pm 0.23$ | 130 M | 1.01 G |
| SPM-CRC [43] | 93.86 | 23 M | 856 M |
| AResNet + WSPM-CRC [43] | 93.60 | 25 M | 727 M |
| Aggregated Deep Fisher Feature [39] | $95.21 \pm 0.50$ | 23 M | 513 M |
| SE-MDPMNet [42] | $92.46 \pm 0.66$ | **5.2 M** | 3.27 G |
| Variable-Weighted Multi-Fusion [61] | 89.1 | 53 M | 334 M |
| Contourlet CNN [46] | $95.54 \pm 0.17$ | 21.6 M | 2.1 G |
| Positional Context Aggregation [49] | $95.98 \pm 0.56$ | 28 M | 8.6 G |
| LCNN–BFF Method [40] | $94.64 \pm 0.12$ | 7 M | **24.6 M** |
| Proposed | **$98.02 \pm 0.10$** | 19 M | 198 M |

Compared with other models, although the number of parameters of our model is not the lowest, it is less than most models. Specifically, we have 13.8 M more parameters than SE-MDPMNet [42], but our accuracy is the highest. In addition, compared with VGG-16-CapsNet [15] and variable-weighted multi-fusion [61], the number of parameters of our model is greatly reduced. Compared with SPM-CRC [43], AResNet + WSPM-CRC [43], aggregated deep Fisher feature [39], Contourlet CNN [46], and positional context aggregation [49], the number of parameters of the proposed model is about 5 M fewer. According to the last column of Table 12, the FLOPs of our method are still fewer than most models. Overall, the proposed model achieves a good balance of parameters complexity and accuracy.

### 3.3.8. Visual Analysis of the Model

To demonstrate the feature extraction ability of our model, two different visual methods are implemented to evaluate our model. First of all, the T-distributed stochastic neighbor embedding visualization algorithm (T-SNE) [71] is selected to visualize feature representations learned by the proposed model. The T-SNE algorithm is a machine learning method for dimension reduction, which generally reduces the high dimensions to 2D space. In this section, the T-SNE algorithm is performed on the AID and UCM datasets, as shown in Figures 18 and 19. It is obvious that our method has brilliant global feature representations and can clearly distinguish various remote sensing scene categories. We believe this is due to the mmsCNN–HMM in our model, which enables our model to extract rich structural feature information. Apart from that, the stacking ensemble learning scheme in the proposed model effectively prevents overfitting while ensuring accuracy.

Subsequently, the class activation map (CAM) method [72] is adopted to evaluate our model. The CAM method can highlight the part of the object with recognition detected by the CNN. In this section, the CAM method is performed on the AID dataset, as shown in Figure 20. We find that the proposed method can accurately find objects and has a wide highlight range. We believe this is because of the shortcut connect mechanism in the mmsCNN that enables our model to have a larger effective receptive field. In addition, the HMM enables our model to further utilize context information.

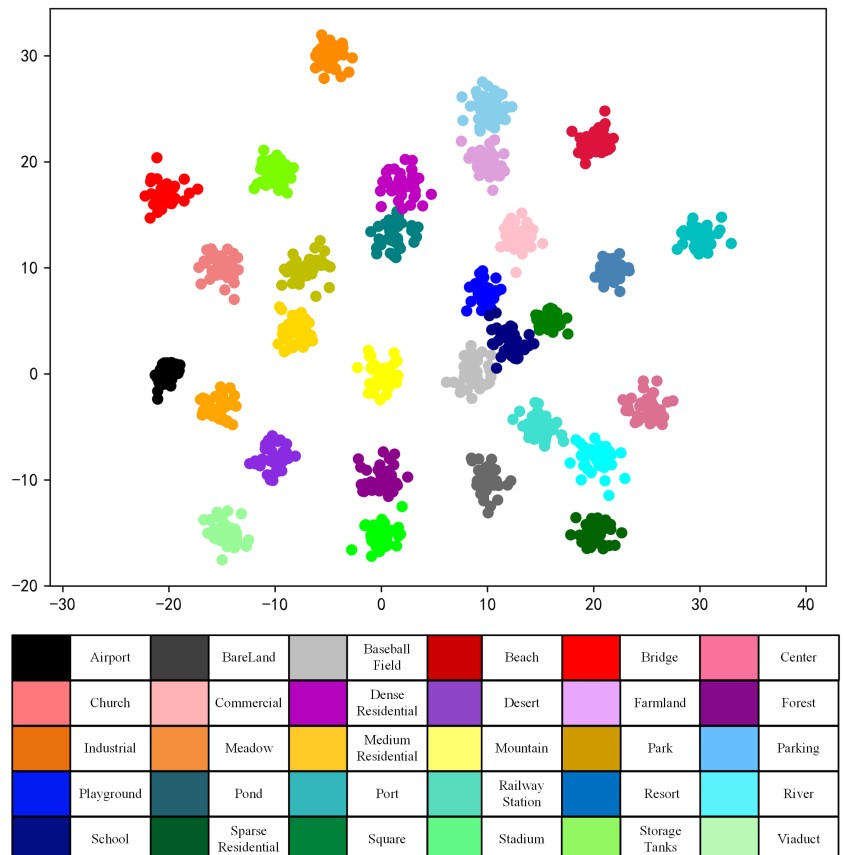

| | | | | | | | | | | | |
|---|---|---|---|---|---|---|---|---|---|---|---|
| ■ | Airport | ■ | BareLand | ■ | Baseball Field | ■ | Beach | ■ | Bridge | ■ | Center |
| ■ | Church | ■ | Commercial | ■ | Dense Residential | ■ | Desert | ■ | Farmland | ■ | Forest |
| ■ | Industrial | ■ | Meadow | ■ | Medium Residential | ■ | Mountain | ■ | Park | ■ | Parking |
| ■ | Playground | ■ | Pond | ■ | Port | ■ | Railway Station | ■ | Resort | ■ | River |
| ■ | School | ■ | Sparse Residential | ■ | Square | ■ | Stadium | ■ | Storage Tanks | ■ | Viaduct |

**Figure 18.** The T-SNE algorithm visualization results of our model on AID dataset.

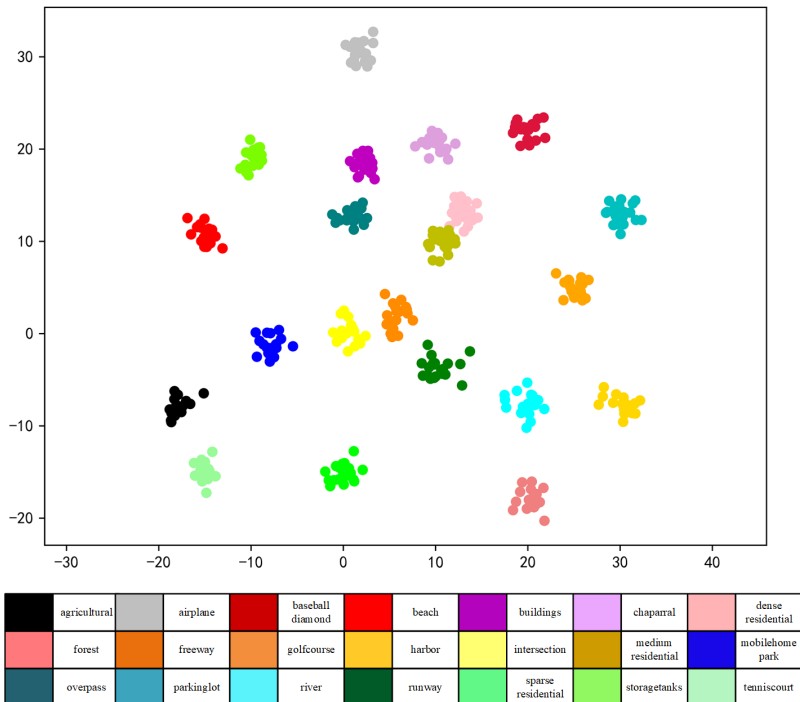

| | | | | | | | | | | | | | |
|---|---|---|---|---|---|---|---|---|---|---|---|---|---|
| ■ | agricultural | ■ | airplane | ■ | baseball diamond | ■ | beach | ■ | buildings | ■ | chaparral | ■ | dense residential |
| ■ | forest | ■ | freeway | ■ | golfcourse | ■ | harbor | ■ | intersection | ■ | medium residential | ■ | mobilehome park |
| ■ | overpass | ■ | parkinglot | ■ | river | ■ | runway | ■ | sparse residential | ■ | storagetanks | ■ | tenniscourt |

**Figure 19.** The T-SNE algorithm visualization results of our model on UCM dataset.

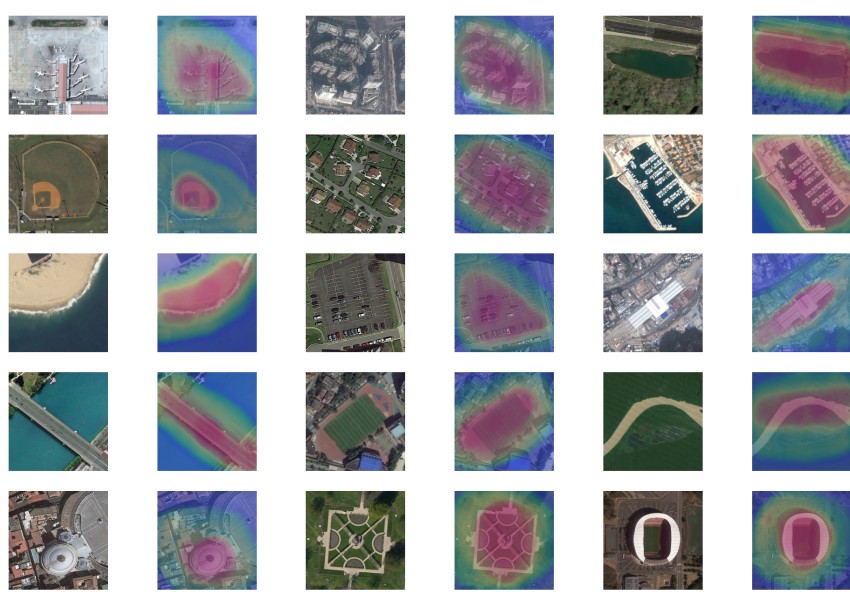

**Figure 20.** The CAM method visualization results of our model on AID dataset.

## 4. Conclusions

In this paper, a new classification approach using an mmsCNN–HMM combined model with stacking ensemble mechanism is proposed to classify remote sensing scene images. First of all, a lightweight mmsCNN with the shortcut connections mechanism is designed to extract multi-scale structural features from remote sensing images, which can avoid high computational complexity and gradient vanishing. Next, an appropriate HMM is designed to mine the context information of the extracted features by mmsCNN, which can obtain richer hidden structure information. For different categories of scene images, the corresponding HMM is trained and all the trained HMMs form an HMM group. Additionally, our method is on the basis of a stacking ensemble learning scheme, in which the prediction generated by the basic learning method is used by the top-level method to generate the final prediction. Specifically, the preliminary prediction results generated by the trained HMM group are fed into the extreme gradient boosting (XGBoost) model to conduct the scene class prediction, which can effectively avoid overfitting while ensuring accuracy. A great deal of experiments were performed on the six most widely used datasets. The numerical experiments verify that the proposed approach shows more important advantage than the advanced approaches. Finally, our future work aims to design a more efficient and lightweight convolutional neural network.

**Author Contributions:** Methodology, X.C.; formal analysis, X.C.; investigation, X.C.; writing—original draft preparation, X.C.; writing—review and editing, X.C. and H.L.; visualization, X.C.; supervision, H.L.; project administration, H.L. All authors have read and agreed to the published version of the manuscript.

**Funding:** This research received no external funding.

**Data Availability Statement:** Datasets relevant to our paper are available online. The code is available at https://github.com/tttttcx/Remote-Sensing-Scene-Image-Classification-Based-on-mmsCNN-HMM-with-Stacking-Ensemble-Model.

**Conflicts of Interest:** The authors declare no conflict of interest.

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
