# Peer review of "Remote Sensing Scene Image Classification Based on mmsCNN–HMM with Stacking Ensemble Model"

_remotesensing, doi:10.3390/rs14174423_

Round 1

Reviewer 1 Report

Good paper

Reviewer 2 Report

Overfitting is a well-known problem in DL-based image classification, especially when it comes to CNNs. Therefore, addressing this problem and trying to solve it could be a useful contribution to the field of HSI RS classification. The authors targeted this problem with a stacking ensemble learning mechanism that integrates multiple models to make decisions together.

Generally speaking, the manuscript is well-structured, the topic is very interesting and to-the-point, the experimental results are adequate (as six datasets are being used), and the language is fine, however, it needs to be improved. The innovation level is not high since the main idea is to combine well-known tools and concepts with minor modifications, however, the outcome is very inviting and that could cover the level of novelty in the submitted work.

In general, I found the submitted work interesting and a good fit to the state-of-the-art, but I would also have some concerns and comments that need to be addressed. I list them here:

Major:

-         Although the computational complexity is one of main concerns of the presented study, I wonder why it is not addressed and discussed sufficiently in the paper. I would significantly suggest adding a sub-section to the manuscript, reporting the complexity of the proposed method in terms of the computational time, number of parameters and so on. Adding new tables to the manuscript for reporting such data could be an option. My concern is about the complexity of the proposed method since the framework seems a bit heavy to me, therefore, I am curious to see how complex the whole pipeline could be. Also, a comparison to the other competing methods in terms of the complexity could really enrich the study.

Minor:

-         That would be very useful if a reference for the AID data is also provided in the caption of Figure 1.

-         Line 38, ‘approaches’ instead of approach

-         Line 96, ‘An appropriate’

-         Although it was new to me, I really liked the way the authors tried to explain the mathematical symbols, at the end of the Introduction section and before stepping through the main part of the article. Not sure if the style is approvable by the MDPI RS journal, but I personally found it interesting and very useful.

-         In caption of Figure 2, feel free to remove the few starting words: ‘The figure shows’.

-         Same comment applies to the caption of Figure 5.

-         Although the HMM has been already discussed and cited adequately within the manuscript, I still suggest adding a reference to the first sentence of section 2.2, when HMM is being introduced within its specific chapter.

-         In caption of Figure 4, it is already obvious from the figure that the length of the HMM is T.

-         Line 227, are these exactly the MOST widely used RS scene datasets?

-         Line 230, it reads: “To guarantee the validity of experiments, every model is performed on the same hyperparameters and equipment.”. Thank you for making sure about a fair assessment. But by ‘every model’, do you mean the proposed model that is being trained/tested each time on one of the 6 datasets? Or you are talking about the other competing models? It the latter is the case, using the same hyperparameters for the other competing models does not necessarily lead to a fair comparison, since they need to be run using their optimum hyperparameters. But if the former is the case, please ignore this comment.

-         Table 3 is concise but great! I just wanted to thank the authors, as they used several other state-of-the-arts in the comparison tables. Same comment for the following tables.

Reviewer 3 Report

To classify remote sensing scene images, a new classification approach that employs a mmsCNN-HMM combined model with stacking ensemble mechanism is proposed in this paper. First and foremost, in order to extract multi-scale structural features from remote sensing images, a lightweight mmsCNN that utilizes the shortcut connections mechanism has been developed. This helps to avoid high computational complexity as well as gradient vanishing. After that, an appropriate HMM is designed to mine the context information of the features that were extracted by mmsCNN. This allows for the acquisition of more detailed information regarding the hidden structure. The corresponding HMM is trained for each distinct category of scene image, and the culmination of all of the trained HMMs is referred to as the HMMgroup. In addition, authors' approach makes use of a stacking ensemble learning scheme, which means that the prediction that is produced by the fundamental learning method is utilized by the top-level method in order to produce the ultimate prediction. To be more specific, the preliminary prediction result that was generated by the trained HMM group is fed into the eXtreme Gradient Boosting (XGBoost) model in order to conduct the scene class prediction. 

Overall, the article seems well written and worthy of publication. I would only ask that the references be expanded a bit to include other more recent studies on the subject.

I suggest a re-reading to eliminate inaccuracies in the formatting, admittedly very few. For example:

Figure 6, Lines 269, 285, 298, 312, 357: Figures, as was correctly done for the others, should also be cited before placement in the main text. 

Reviewer 4 Report

In the paper, a method for remote sensing image scene classification is proposed. Comments:
1.    The used “invalid layers” should be explained (page  6), as they cannot be found in the literature. Otherwise, please use a different naming as invalid indicates something unimportant or faulty.
2.    Figure 4 can be safely removed as it presents common knowledge.
3.    Hidden Markov Models are often used to capture time-related dependencies between features in their sequences. Please elaborate on the similarities and impact of the way the image can be transformed into a sequence of features on the performance of the method. Is the precedence of features important? The usage of HMMs should be better explained. Why is it suitable for the task? Given a vector of features from mmsCNN many alternative solutions could be used instead. Is this a vector or a sequence of features? The sequence of features resembles the time series of some sensor data and it would be well recognized by the HMMs but there are better methods that could classify testing data based on vectors of features. Note that, there are better classifiers for sequences than classical HMMs. The mined “context information” should be shared with readers. A solid investigation is expected here.
4.    Since the “lightweight structure” is proposed, the computational/resource usage/complexity gains should be discussed.
5.    The best results in tables should be highlighted (boldface).
6.    The need for stacking is not supported experimentally. Other ensembles should be investigated.
7.    The paper lacks the ablation study. What would happen if Xgboost was not used?
8.    The results cannot be replicated by a reader. The paper should contain a link to a source code of the method, ensuring the repeatability of the results. 
9.    The requires thorough proofreading and fixing grammar errors (e.g., Abstract, first sentence: “an significant”), or addressing informal language (e.g., page 2: “has become a hot word”).

Round 2

Reviewer 2 Report

I would like to thank the authors for carefully addressing my comments, especially my concerns about the lack of information about the complexity of the proposed method. I have no further comments or remarks at this step.

Reviewer 4 Report

The revision improved the paper. Further comments:
1. Please discuss the importance of the features’ priority exploited by the HMMs using a clear example.   
2. The method is complex and the results cannot be replicated by a reader. The paper should contain at least a link to a webpage (e.g., Github) where it will be placed after the paper is accepted (mandatory point).
